# GC-FID-MS Based Metabolomics to Access Plum Brandy Quality

**DOI:** 10.3390/molecules26051391

**Published:** 2021-03-05

**Authors:** Stefan Ivanović, Katarina Simić, Vele Tešević, Ljubodrag Vujisić, Marko Ljekočević, Dejan Gođevac

**Affiliations:** 1National Institute of the Republic of Serbia, Institute of Chemistry, Technology and Metallurgy, University of Belgrade, Njegoševa 12, 11000 Belgrade, Serbia; stefan.ivanovic@ihtm.bg.ac.rs (S.I.); katarina.simic@ihtm.bg.ac.rs (K.S.); 2Faculty of Chemistry, University of Belgrade, Studentski trg 12-16, 11000 Belgrade, Serbia; vtesevic@chem.bg.ac.rs (V.T.); ljubaw@chem.bg.ac.rs (L.V.); 3Zarić Distillery, Maksima Markovića 42, 31260 Kosjerić, Serbia; marko.lekocevic@gmail.com

**Keywords:** plum brandy, metabolomics, GC-FID-MS

## Abstract

Plum brandy (Slivovitz (en); Šljivovica(sr)) is an alcoholic beverage that is increasingly consumed all over the world. Its quality assessment has become of great importance. In our study, the main volatiles and aroma compounds of 108 non-aged plum brandies originating from three plum cultivars, and fermented using different conditions, were investigated. The chemical profiles obtained after two-step GC-FID-MS analysis were subjected to multivariate data analysis to reveal the peculiarity in different cultivars and fermentation process. Correlation of plum brandy chemical composition with its sensory characteristics obtained by expert commission was also performed. The utilization of PCA and OPLS-DA multivariate analysis methods on GC-FID-MS, enabled discrimination of brandy samples based on differences in plum varieties, pH of plum mash, and addition of selected yeast or enzymes during fermentation. The correlation of brandy GC-FID-MS profiles with their sensory properties was achieved by OPLS multivariate analysis. Proposed workflow confirmed the potential of GC-FID-MS in combination with multivariate data analysis that can be applied to assess the plum brandy quality.

## 1. Introduction

Plum brandy (Slivovitz (en); Šljivovica(sr)) is produced in Eastern and Central Europe, and consumed worldwide. The quality of plum brandy depends on many factors, such as: environmental characteristics of plum cultivation, characteristics of plum cultivars, and technological characteristics of the production process [1]. Revenue in the brandy segment was USD 63,620 million in 2020 worldwide, and the market is expected to exhibit a strong 6.9% compound annual growth rate over the forecast period from 2020 to 2023 [2]. Beside ethanol, plum brandy contains a complex mixture of chemical compounds formed by plum and yeast metabolism during the distillation process.

Domestic or European plum (*Prunus domestica* L.) is the most represented and traditionally the most important fruit species in Serbia. In the world production of plums, the Republic of Serbia was ranked second in Europe in 2019 and third in the world, behind China and Romania, with a production of 558,930 t [3]. The most popular plum cultivars in Serbia are Požegača, Čačanska rodna, Stanley, Čačanska lepotica, Čačanska najbolja, and Čačanska rana [4].

Plum fruits may be consumed fresh, dried, or prepared into preserves, compotes, mousse, pulp, candied fruit, frozen fruit, jams, and jelly products [5]. The largest amount of plum fruits produced in Serbia (more than 75%) is processed into brandy [6].

Serbia has a long tradition of manufacturing natural fruit brandies and plum spirits called Rakija or Šljivovica are produced as a national drink.

In the production of plum spirits with distinctive aromatic characteristics, old widespread plum cultivars (such as plum cultivars Požegača (PZ)) have been traditionally used. Additionally, some previously rarely used and autochthonous cultivars have been used more intensively as a raw material in the production of spirits. There are a number of old native cultivars, such as Crvena ranka (CR) (*P. domestica* L.) and Trnovača (TR) (*Prunus insititia* L.), which are suitable for high quality brandies. CR plum (also known as Darosavka, Šumadinka, Crvenjača, or Ranošljiva) is an autochthonous variety of Serbia [7].

Traditional fruit brandy production still mostly uses a simple distillation pot. In recent years, column distillation has slowly entered the production of spirits in small distilleries and two different types of distillation equipment are now commonly used in the production of fruit spirits: the Charentais alembic pot (French style) and the batch distillation column (German style) [8].

Apart from water and ethanol as the main constituents, brandy also contains a number of other components, the concentration of which is mostly dependent upon the cultivar, production still, and differences in microflora during fermentation. The quality of the fruit spirits depends on the concentration and the relationship of the individual volatile compounds. He et al. analyzed a traditional Chinese brandy using solid head microextraction and GC-MS. The obtained volatile profiles were then analyzed by OPLS-DA for classification of the samples according to their aroma and region [9]. A number of studies on the compositions of volatile compounds in both stone and pome fruit spirits, such as plum brandy, apricot brandy, mirabelle brandy, and Cornelian cherry brandy have been reported [9,10,11,12,13,14,15,16].

Volatile compounds are one of the main characteristics that determine a brandy’s organoleptic quality and style. This is the result of the contribution of hundreds of volatile compounds, including higher alcohols, esters, acids, aldehydes, ketones, terpenes, norisoprenoids, and volatile phenols. These compounds can be classified into four groups: primary aromatic compounds (whose entire aroma appears exactly as in the fruit during ripening); secondary aromatic components (formed during alcoholic fermentation); tertiary aromatic compounds (formed during the distillation process); and quaternary aromatic compounds (formed during the maturation process) [11,12,13,14,15,16].

In most cases, the producers make fruit brandies in the traditional way without using selected yeast strains, enzymes, or other agents. However, in order to obtain good quality fruit brandy, many producers use selected yeast strains.

The work reported by Satora and Tuszyński [17] showed that the samples obtained after spontaneous fermentation were distinguished by a high content of acetoin, ethyl acetate, and total esters, accompanied by a low level of methanol and fusel alcohols. They have also found that non-*Saccharomyces* yeasts were responsible for higher concentrations of esters and methanol, while *S. cerevisiae* strains resulted in increased levels of higher alcohols. In comparison with isolated indigenous strains of *S. cerevisiae* synthesized relatively low amounts of higher alcohols in regard to commercial cultures [11].

The quality of plum brandy is a complex concept. In addition to its chemical composition, the utmost importance is attributed to its sensory characteristics as one of the most important factors of quality. The results of sensory evaluations together with the results of chemical analysis may be used to make a final judgement on the quality of plum brandy only after their processing by adequate mathematical-statistical methods [18].

M. Jakubíková et al. used synchronous fluorescence, UV−Vis, and near infrared (NIR) spectroscopy coupled with chemometric methods to discriminate samples of high-quality plum brandies of different varietal origins (*Prunus domestica* L.) [19]. Bajer et al. analysed volatile compounds profiles of various fruit spirits (plum, mirabelle, apricot, pear, and apple spirits) by method of headspace solid-phase microextraction coupled to gas chromatography with flame ionization detection (HS-SPME/GC-FID) and obtained chromatographic data used for authentication of fruit spirits [20]. Based on the results obtained by HPLC–FLD analysis and using chemometric methods (principal component analysis (PCA) and linear discriminant analysis LDA), Jakubíková et al. detected and quantified two volatile phenols (eugenol, 4-ethylphenol) and three anisoles (4-vinylanisole, 4-allylanisole, 4-propenylanisole) in plum brandies with different origins. Moreover, quantitative profiles of the eugenol, 4-vinylanisole, and 4-ethylphenol showed high diversity between summer and autumn plum brandy samples [21].

The chemical complexity and high alcohol content of fruit brandy make sensory evaluation of the product very challenging. Moreover, sensory evaluation of brandy must be strictly controlled to ensure valid results.

In our study, GC-FID-MS matabolomics was employed to discover fine differences in non-aged plum brandies chemical composition establishing desirable and undesirable metabolites that make differences between plum brandies of different quality. The main volatiles and aroma compounds of 108 plum brandies originating from three plum cultivars, and fermented using different conditions, were investigated. The chemical profiles obtained after two-step GC-FID-MS analysis were subjected to multivariate data analysis to reveal the peculiarity in different cultivars and fermentation process.

Finally, correlation of plum brandy chemical composition with its sensory characteristics obtained by expert commission was performed.

## 2. Results and Discussion

### 2.1. GC-FID-MS Profiling of Plum Brandies

To follow the variability of plum brandies composition in regard to plum cultivars and using different technological process prior fermentation, the qualitatively and quantitatively measurements of all molecules present therein is needed. This approach is well known in plant or human metabolomics studies, where GC-FID-MS in combination with multivariate data analysis is the method of choice in case of volatile compounds [22]. In case of plum brandies, it is an even more suitable method because both brandies and gas chromatography share the same technology—the distillation process.

The 108 plum brandies were produced from three plum cultivars using different fermentation conditions: two different pH adjustment of plum mash, addition of yeast Lalvin QA23, addition of enzymes Lallzyme BETA™ and Lallzyme CUVÉE BLANC™.

In total, 89 compounds were identified and taken into account for aroma metabolites profiling.

### 2.2. Multivariate Models Creation

All the 89 volatiles and aroma metabolites concentrations obtained from flame ionization detector (FID) areas for 108 samples were subjected to multivariate data analysis. Firstly, principal component analysis (PCA) as an unsupervised variable reduction technique to develop smaller number of novel variables that will account for most of the variation in the observed variables was performed. It resulted in 13 principal components (PCs) model explaining 83.1% of the total data variance. The optimal model dimensionality is obtained after 7-fold internal cross-validation. If a new PC improved the predictive power compared with preceding PC, the new PC is retained in the model. Based on PCA score plot of the first two PCs (Figure 1a), three groups of samples were separated according to the plum cultivars. Interestingly, the samples were also separated along PC5 due to different pH value of plum mash prior the fermentation (Figure 1b).

Next, orthogonal partial least squares to latent structures-discriminant analysis (OPLS-DA) was performed. In this supervised technique, novel variables will account for maximum separation between predefined classes. An additional advantage of the orthogonal model is the facilitated interpretation due to separation of the systematic variation of the variables into two parts: one linearly related to class information and one orthogonal to the class information [23]. Thus, OPLS-DA is suitable for finding variables having the greatest discriminatory power between classes. Two OPLS-DA models were created containing CG-MS-FID data of brandies produced from different cultivars: CR versus PZ and TR versus PZ. To investigate the influence of different technological process prior to fermentation, more OPLS-DA was performed: pH value of plum mash (pH 3 versus pH 3.5), addition of yeast Lalvin QA23 versus natural occurring yeast, addition of enzyme Lallzyme BETA™ versus no addition of enzyme, and addition of enzyme Lallzyme CUVÉE BLANC™ versus no addition of enzyme.

The quality of the obtained models was assessed by goodness of fit (R^2^) indicating how well the variation of variables is explained using the predictive components and predictive ability of the model (Q^2^) indicating how well the model predicts new data, as estimated by cross validation. In the five OPLS-DA models concerning different cultivars, pH of plum mash, yeast and Lallzyme BETA™ addition, R^2^ and Q^2^ values over 0.9 and close to 1 (maximum value) indicated remarkable goodness of fit, and predictivity (Table 1).

The OPLS-DA models were validated in two ways. Firstly, in permutation tests the R^2^ and Q^2^ values of the original models were compared with the R^2^ and Q^2^ values of several models based on data where the order of class information has been randomly permuted. The adequate results were obtained for all the five OPLS-DA models since regressions of Q^2^ lines intersected the vertical axis at below zero, and all Q^2^ and R^2^ values of permuted Y vectors were lower than the original ones. Further validation was performed by CV-ANOVA, where the significance of the five OPLS-DA models was clearly shown with *p* values far less than 0.05, except the model related to addition of Lallzyme CUVÉE BLANC™ (Table 1). In the misclassification tables (Appendix A), performance metrics for classifications of the OPLS-DA models are summarized.

The score plots of five OPLS-DA models are depicted in Figure 1c–g. This indicated good separation between classes along to the predictive components.

Finally, orthogonal partial least squares to latent structures (OPLS) analysis was applied to correlate brandy composition with its sensory properties. The jointed volatiles and aroma metabolites concentrations obtained from FID areas were used as X variables, while average grades from sensory analysis performed by expert commission were used as Y variables. The goodness of fit and predictive ability were fairly good according to Q^2^ and R^2^ values. According to CV-ANOVA, the OPLS model was significant with *p* < 0.05 (Table 1). This is also in accordance with the results of permutation test. According to the score plots of the OPLS model, brandy samples with lower grades (15−16) were clearly separated from those with higher grades (17–18.5). The area of grades 17–18.5 were slightly separated (Figure 1h).

The selection of the most influential variables in OPLS and OPLS-DA models was based on variable influence on projection scores of the predictive components (VIPpred). Variables with the VIPpred score above 1.4 were considered as important for the correlation or separation.

### 2.3. Cultivar Influence on Brandy Profiles

Since two OPLS-DA models containing data of brandies produced from three different cultivars, shared and unique structure plot (SUS-plot) was used to reveal the metabolites specific for each cultivar. The SUS-Plot represents a scatter plot of the loadings scaled as a correlation coefficient (p(corr) (1)) from two separate OPLS-DA models [23] (Figure 2). The negative values of loadings coresponded to the most influential variables responsible for separation of CR and TR in models M2 and M3, respectively. Such loadings for PZ important variables exhibited positive values in both models.

Thus, metabolites **11**–**17** found close to the extreme negative end of the X axis in the SUS-plot having CR/PZ p(corr)(1) value close to −1 and TR/PZ p(corr)(1) value close to zero were discriminating for CR cultivar. Similarly, the metabolites **4**–**10** with both p(corr)(1) values close to 1 were discriminating for PZ cultivar. The metabolites (**1**–**3**) found in the upper left corner were discriminating for CR and PZ relative to TR. In the same manner, metabolites (**18**–**20**) in the lower left corner of the SUS-Plot were discriminating for TR and CR relative to PZ.

The impact of cultivars on the volatile organic compounds composition of plum brandy has been studied by Vyviurska et al. It was found that plum brandies distilled by the same technology from 25 plum cultivars could be distinguished by two-dimensional gas chromatographic analysis and sensory evaluation. The main differences were observed in the presence of unsaturated fusel alcohols (3-methyl-3-buten-1-ol, trans-3-hexenol), unsaturated aldehydes (2-butenal, 2-nonenal), monoterpene derivatives (linalool acetate, geraniol acetate) and lactones, which were mainly detected at the trace level [24].

Ethyl esters present quantitatively the largest group of the aroma compounds in the plum distillates. Esters are formed during alcoholic fermentation via yeast metabolism and qualitatively present the major class of flavor compounds in distillates [12].

Our results showed that fatty acid ethyl esters **6**, **8**, **12**, and **14** discriminated the plum brandies obtained from cultivar CR. Ethyl butyrate (**7**) was discriminating ester for plum brandies obtained from cultivar PZ. Ethyl stearate discriminated plum brandies obtained from cultivar TR and CR, while ethyl laurate and ethyl benzoate discriminated brandies from CR and PZ. The esters contents in the plum brandy were in agreement with the previously reported data [10].

Higher alcohols are common constituents in alcoholic beverages and are formed in small amounts by yeast, from sugars and from amino acids metabolism during the alcoholic fermentation process. The most common fusel alcohols in distilled spirits include: 2-butanol, *i*-butanol, 1-butanol, 1-propanol, and amyl alcohols [17]. In our work, 1-butanol discriminating brandy was obtained from CR, while *i*-butanol and (Z)-3-Hexen-1-ol were discriminating in the distillates obtained from PZ cultivar.

Diethyl acetals derived from the reaction between the corresponding aldehydes and ethanol, probably during the distillation process [25].

1,1-Diethoxyhexane was a discriminating component for the cultivar PZ and 1,1-diethoxybutane for PZ and CV cultivar.

### 2.4. Fermentation Process Influence on Brandy Profiles

The volatile profile, taste and flavor as well as yield of plum brandy is strongly dependent on the unique aromatic profile of plum and the yeast present on the surface of the fruit, as well as the yeast used for nutrition [26]. The majority flavor compounds are formed during fermentation by the yeast. These compounds include volatile organic acids, alcohols, aldehydes, and esters. The production and the amount of these compounds found in the fruit brandy are yeast strain dependent.

Spontaneous fermentation of plum musts involves a sequential succession of yeasts. During the first phase of fermentation *Kloeckera apiculata* and *Candida pulcherrima* are dominated strains. However, as fermentation progresses, these non-*Saccharomyces* species are naturally substituted by *Saccharomyces* ones, because the latter are good fermenters and possess high alcohol tolerance [17].

In our study, the samples after spontaneous fermentation were distinguished by a high content of fatty acid ethyl esters (**32**, **34**–**38**) and nonyl acetate (**33**). Selected yeast was responsible for higher concentrations of metabolites **23**, **27** and **39**–**44** (Figure 4).

Satora and Tuszyński have described plum must fermentations by different yeasts isolated from fresh blue plum fruits (*Aureobasidium* sp.) and spontaneously fermenting plum musts (*Kloeckera apiculata* and *Saccharomyces cerevisiae*), as well as commercial wine and distillery strains non-*Saccharomyces* yeasts. They found that samples after spontaneous fermentation were distinguished by a high content of acetoin, ethyl acetate, and total esters, accompanied by a low level of methanol and fusel alcohols. Non-*Saccharomyces* yeasts were responsible for higher concentrations of esters and methanol, while *S. cerevisiae* strains resulted in increased levels of higher alcohols. It was also found that isolated indigenous strains of *S. cerevisiae* synthesized relatively low amounts of higher alcohols compared to commercial cultures [17]. Urošević et al. evaluated the influence of yeast and nutrients on the quality of apricot brandy using five yeast strains *S. cerevisiae* and *S. bayanus*. The best results were obtained with yeast strain *S. cerevisiae* ex r.f. *bayanus* and yeast strain *S. cerevisiae* ex ph.r. *bayanus*, which gave a high content of linalool. The control sample with no nutrients and selected yeast gave a distillate that was evaluated as having the worst quality with higher concentration of ethyl acetate, 1-butanol, 1-hexanol, and amyl alcohols [27].

Among many constituent of plum are organic acids, which are of great importance for alcohol fermentation chemical processes. Plum fruits do not contain higher amounts of acids and their pH is over 3.5, in most cases [28]. pH value affects the growth and fermentation rate of yeast. Our results have shown that pH value considerably influenced the composition of fermentation products. Benzene derivatives (**2**, **19**, **21**, **23**–**25**) were a dominate group of volatile compounds in brandies obtained on pH = 3.5. Decreasing of the pH value to pH = 3.0 led to increased levels of acetals (**27**, **30**), monoterpenes (**29**, **31**) and esters **26**, **28**, and **32**.

In alcoholic fermentation, enzymes act as catalysts for carbohydrate decomposition reactions and the formation of specific compounds. The use of enzyme preparations also has a significant impact on the quality of plum brandy [27].

Treatment of fermentation broths with pectic enzyme complex Lallzyme BETA™ containing pectinases, beta-glucosidase, rhamnosidase, apiosidase, and arabinofuranosidase provoked increased yield of ethyl esters (**17**, **28**, **45**, **48**), acetates (**33**, **46**, **47**, **49**), eugenole (**24**) and propionic acid (**50**). On the other hand, the treatment with Lallzyme CUVÉE BLANC™, which contains mainly pectinases, did not show any effect on brandy composition.

### 2.5. Correlation of Brandies Profiles with Their Sensory Properties

In general, the most common volatiles that contribute to the flavor and/or aroma profile of fruit brandies can be broadly categorized into several main groups: higher alcohols, esters, acetates, organic acids, etc. Nevertheless, there are many minor volatile and nonvolatile components such as aldehydes, ketones, acetals, lactones, terpenes, phenols, and glycerol, which contribute to the full bouquet of flavors, aroma, and mouthfeel of brandies.

The most of the straight-chain alcohols have a strong pungent smell. At low concentrations, they contribute to the aromatic complexity but at higher levels are characterized by penetrating odors, which mask the aromatic finesse. In distilled spirits, such as brandies, rum, and whisky, fusel alcohols provide most of their common aromatic character [29]. As seen in Figure 5, higher alcohols propanol (**39**), amyl alcohols (**51**) and 1-hexanol (**52**) are metabolites that contribute favorable sensorial properties of plum brandy. 1-Hexanol (**52**) usually has a positive influence on the aroma of a distillate when its concentration reaches 20 mg/L [30], but it also will negatively affect the flavor of brandies by imparting a grass-like note, when the concentration exceeds 100 mg/L [11]. The amyl alcohols/isobutanol and isobutanol/propanol ratios are used in some alcoholic beverages such as whisky as a quality index and should be greater than unity [31]. Studies by Satora et al. found that the plum brandies contained more propanol than isobutanol, without any influence on quality of the beverage [13].

Lactones (**18**, **54**) were the second dominant class of desirable compounds in plum brandies. These compounds, particularly γ-lactones, are important compounds in terms of their contribution to the aroma and, in general, present fruity odor descriptors. γ-Lactones are important flavor and aroma constituents and occur in fruits such as apricot, peach, plum, and strawberry. Horvat et al. [14] found that lactones were also a major class of volatile compounds in southeastern American plums. Studies reported by Gomez et al. designate γ-dodecalactone as the major lactone in Japanese plum (*P. salicina*) [15].

Isopentenyl palmitate (**53**) was also obtained in plum brandy. This ester is rare in fruit brandy, but may be responsible for the apple, pineapple, and banana whiskey odor [32].

Acetates (**33**, **46**, **47**, **55,** and **57**), esters (**7**, **28**, **59**), acetals (**4**, **60**, **56**), octanal (**58**), and phenyl ethyl alcohol were marked as undesirable molecules, but this should be understood conditionally, because for such a complex aroma mixture, the question of exact concentration and synergistic effect are of great importance.

Ethyl acetate (**47**) is the most common and typical ester in fruit brandy. In small concentrations, they have floral notes but at high concentration, they can be very repulsive, exhibiting the odor of solvent [33].

## 3. Materials and Methods

### 3.1. Materials

Methanol, ethanol, 1-propanol, acetaldehyde, ethyl-acetate, *iso*-butanol, 1-butanol, isoamyl alcohol, 1-hexanol, dichloromethane, 4-methyl-1-pentanol and methyl 10-undecenoate and formic acid of were purchased from Sigma–Aldrich (Steinheim, Germany). Anhydrous magnesium-sulfate was obtained from Merck (Darmstadt, Germany). All chemicals were p.a. purity. Commercial enzymes β Lallzyme™, Lallzyme Cuvee Blanc™, and yeast Lalvin QA23™ were supplied from Lallemand (Montreal, QC, Canada).

### 3.2. Plum Sample Collection

The plum fruits were collected at the stage of technological maturity near the town Kosjerić (Western Serbia) in autumn 2017. Three cultivars were collected: Trnovača, Crvena ranka, and Požegača; taking care that the fruits did not contain any impurities (e.g., soil, leaves, branches) and without rotten or damaged fruits in order to avoid the influence of factors that are not the subject of these experiments. Each variety was harvested from the same localities and plantations, which eliminates the influence of the difference in results due to different microclimates, soil composition, or agrotechnics applied in orchards.

### 3.3. Brandy Production

The amount of mashed fruits without stones in each experiment was 45 kg. A total of twelve experiments in triplicate were set for each cultivar (Appendix A). The addition of enzymes β Lallzyme™, Lallzyme Cuvee Blanc™, yeast Lalvin QA23™, and pH value was varied. Concentration of enzyme β Lallazyme™ was 0.05 g/kg in experiments II, V, VIII, and XI. The concentration of Lallzyme Cuvee Blanc™ was 0.02 g/kg in experiments III, VI, IX, XII; and concentration of yeast Lalvin QA23™ was 0.22 g/kg in experiments VII–XII. In experiments IV–VI, and X–XII the pH value was adjusted with metatartaric acid to 3.0, or 3.5. Experiment I was control without the addition of enzymes or yeast. Fermentations were conducted in 60 L plastic containers. When fermentation was done all samples were distilled. Double distillation was conducted in a 15 L copper pot. The first distillate was subjected to a second distillation, and three fractions were collected: the first (“head”), the second (“heart”), and the third (“tail”). The “head” fraction was collected until volume reach 1.5% of total volume of the first distillate. The “heart” fraction was collected until the distillate reached a value of 65% (*v*/*v*) ethanol. The obtained distillates were diluted with demineralized water to the strength of 45% (*v*/*v*). Total of 108 plum brandies, originated from three plum cultivars, were manufactured using different fermentation conditions.

### 3.4. Sample Preparation

A total of 108 plum brandies, originated from three plum cultivars, were manufactured using different fermentation conditions. Main volatiles and aroma compounds in the 108 manufactured plum brandies were determined by GC-FID and GC-FID-MS analysis, respectively.

Sample preparation for determination of main volatiles (acetaldehyde, ethyl-acetate, methanol, 1-propanol, *i*-butanol, 1-butanol, isoamyl alcohol, and 1-hexanol) in the plum brandies by GC-FID was carried out by adding 3 µL of 4-methyl-1-pentanol as an internal standard to 5 mL of brandy. A mixture of authentic standards is made by adding 3 µL of each compound to 5 mL of 40% ethanol [12].

The extraction of aroma compounds for GC-FID–MS analysis was started by diluting an aliquot of 100 mL of plum beverage with 100 mL distilled water, followed by adding of 15 mL of dichloromethane, 1 mL of internal standard (methyl ester of 10-undecenoic acid, 1.8 mg/mL in dichloromethane), and continuously extracted on vortex for 3 min. The dichloromethane extract was dried over anhydrous magnesium sulfate, and concentrated under nitrogen flow to final volume of 1.5 mL.

### 3.5. GC-FID and GC-FID-MS Analysis

The GC–FID analysis of main volatiles were conducting on Agilent 7890A gas chromatograph (Agilent Technologies, Santa Clara, CA, USA) equipped with polar HP-INNOWax capillary column (30 m × 0.32 mm, 0.25 μm film thickness). The analyses were performed in split mode 15:1 with helium as carrier gas at 15.74 psi in constant pressure mode. The injection volume was 1 µL, the injector temperature was 220 °C, column temperature was programmed linearly in the range of 40–90 °C at rate of 5 °C/min with initial 10 min hold, then temperature increased to 240 °C at rate of 50 °C with final 10 min hold. The FID temperature was 300 °C.

The GC–FID-MS analysis of aroma compounds were analyzed on Agilent 7890A GC system (Agilent Technologies, Santa Clara, CA, USA) equipped with a 5975C mass selective detector (MSD) and a FID connected by capillary flow technology through a two-way splitter. A non-polar HP-5MSI capillary column (30 m × 0.25 mm, 0.25 μm film thickness) was used. Column temperature was programmed linearly in the range of 60 °C to 270 °C at a rate of 3 °C/min, then at a rate of 20 °C/min to 310 °C with final 8 min hold. Helium was used as carrier, auxiliary and make up gas; inlet pressure was constant at 19.7 psi (flow 1.0 mL/min at 210 °C), auxiliary pressure was 3.8 psi and FID make up flow was 25 mL/min. FID temperature was 300 °C, split ratio was 5:1 and injection volume was 1 μL for all analysis. Mass spectra obtained by electron ionization with 70 eV at 200 °C. Quadrupole temperature was set to 150 °C and MS range was 40–550 amu. Transfer line temperature was 315 °C.

### 3.6. Data Processing

Identification of main volatiles were done by comparison of retention times of authentic standards in GC-FID chromatograms [10]. Concentrations of these components were calculated, using the following equation:

C_i_ = (A_i_/A_s_) × C_s_ × RF_i_(1)
where: C_i_ is the concentration of the component to be determinedA_i_—the area below the peak of the component of unknown concentrationA_s_—area below peak standardC_s_—concentration standardRF_i_—relative response factor

Library search and mass spectral deconvolution and extraction of aroma compounds were performed using the MSD ChemStation software, version E02.02 (Agilent Technologies, Santa Clara, CA, USA), the NIST AMDIS (Automated Mass Spectral Deconvolution and Identification System) software, version 2.70, and the commercially available Adams04, NIST17, and Wiley07 libraries containing approximately 500,000 spectra. The concentration of the aroma compounds was determined using the peak area of internal standard methyl ester of 10-undecenoic acid, and calculated using the following equation:

C_i_ = (A_i_/A_s_) × C_s_ × DF_i_(2)
where: C_i_ is the concentration of the component to be determinedA_i_—the area below the peak of the component of unknown concentrationA_s_—area below peak standardC_s_—concentration standardDF_i_—dilution factor

The obtained concentrations of main volatiles and aroma compounds were merged in one table, and subjected to multivariate data analysis. Principal component analysis (PCA), orthogonal partial least squares to latent structures (OPLS), and orthogonal partial least squares to latent structures-discriminant analysis (OPLS-DA) methods were performed with SIMCA software (version 15, Sartorius, Göttingen, Germany). The data were mean centered, and scaled to unit variance.

### 3.7. Sensory Analysis of Brandies

Sensory analysis was performed by Buxbaum modified method with a maximum of 20 points [18,28].

The sensory evaluation of brandies was performed in three repetitions. The evaluations were performed by three verified evaluators. The evaluation was anonymous, according to the so-called point-type system. The main quality parameters were evaluated: color, clarity, typicality, smell, and taste.

## 4. Conclusions

The GC-FID-MS-based metabolomics approach was used for the detection of peculiarities in different cultivars and fermentation process, as well as in brandy sensory characteristics. Furthermore, utilization of PCA and OPLS-DA multivariate analysis methods on GC-FID-MS data revealed metabolites important for discrimination between brandies produced under different fermentation conditions (differences in plum varieties, pH of plum mash, and addition of selected yeast or enzymes during fermentation).

Correlation of brandy GC-FID-MS profiles with their sensory properties achieved by OPLS multivariate data analysis. In total, six desirable and 13 undesirable metabolites revealed to support fine tuning between non-aged plum brandies of different quality.

Proposed workflow confirmed the potential of GC-FID-MS in combination with multivariate data analysis that can be applied to assess brandy quality and to understand the chemistry behind the extraordinary plum brandy quality.

Though the authors studied brandies from three local plum cultivars, it is expected that the results and multivariate models could be extended to other stone fruit spirits.

## Figures and Tables

**Figure 1 molecules-26-01391-f001:**
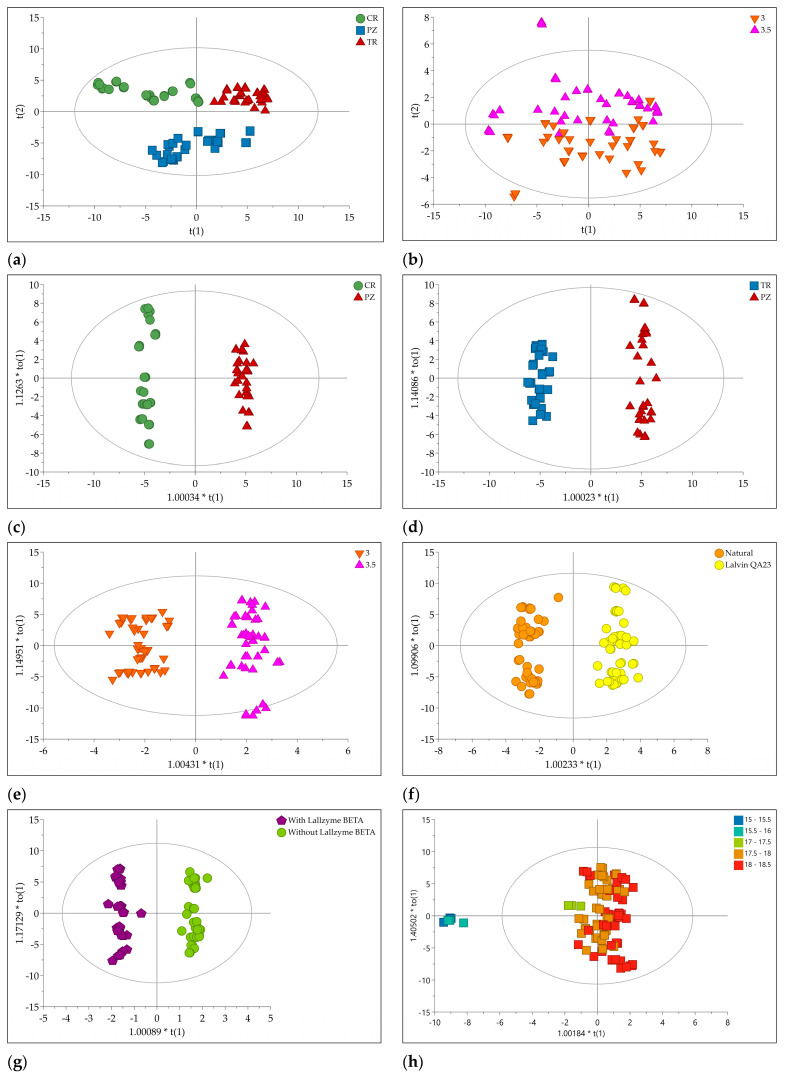
(**a**) PCA score plot (PC1 versus PC2) of all studied samples. The scores are colored according to the cultivars: CR—Crvena ranka, PZ—Požegača, TR—Trnovača; (**b**) PCA score plot (PC1 versus PC5) of all studied samples The scores are colored according to the pH of plum mash; (**c**) OPLS-DA score plot comprising cultivars CR versus PZ; (**d**) OPLS-DA score plot comprising cultivars TR versus PZ; (**e**) OPLS-DA score plot comprising different pH of plum mash, pH 3.0 versus pH 3.5; (**f**) OPLS-DA score plot comprising different yeast, natural versus Lalvin QA23™; (**g**) OPLS-DA score plot comprising enzyme addition, no addition versus Lallzyme BETA™; (**h**) OPLS score plot comprising correlation of brandy composition with its sensory properties. The scores are colored according to the grades of sensory evaluation.

**Figure 2 molecules-26-01391-f002:**
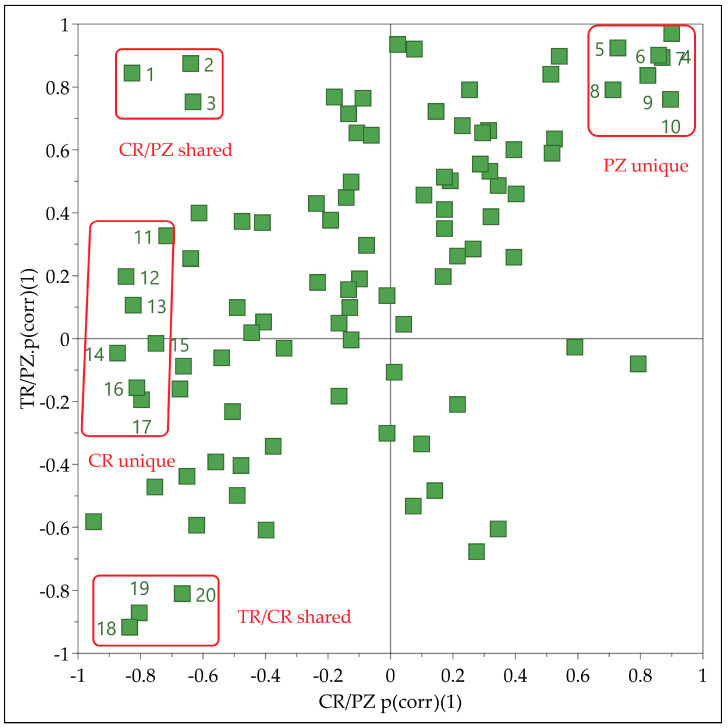
Shared and unique structure plot (SUS-plot) connecting two OPLS-DA models comprising cultivars CR versus PZ (M2) and TR versus PZ (M3). The most influential variables are marked with numbers corresponding to the metabolites depicted in Figure 3.

**Figure 3 molecules-26-01391-f003:**
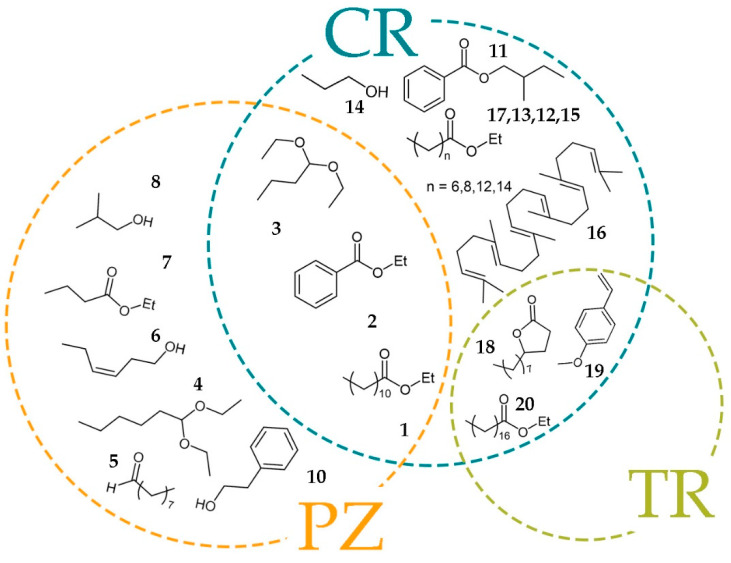
Metabolites important for discrimination between brandies produced from different cultivars.

**Figure 4 molecules-26-01391-f004:**
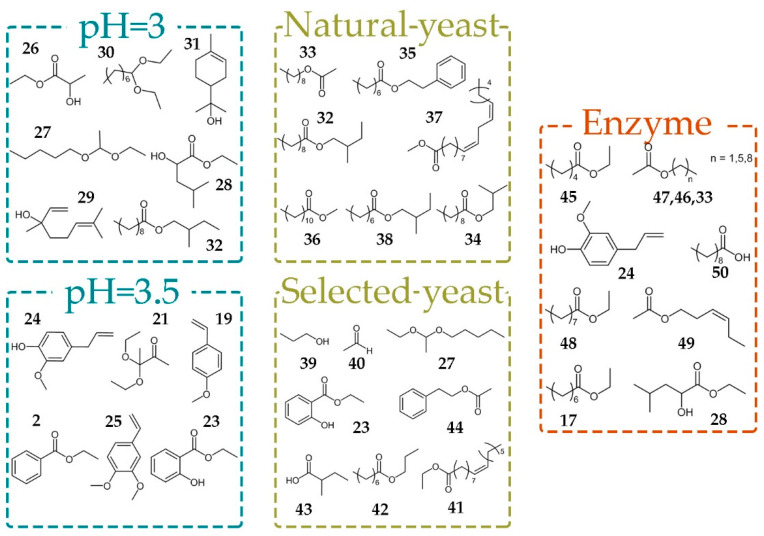
Metabolites important for discrimination between brandies produced under different fermentation conditions.

**Figure 5 molecules-26-01391-f005:**
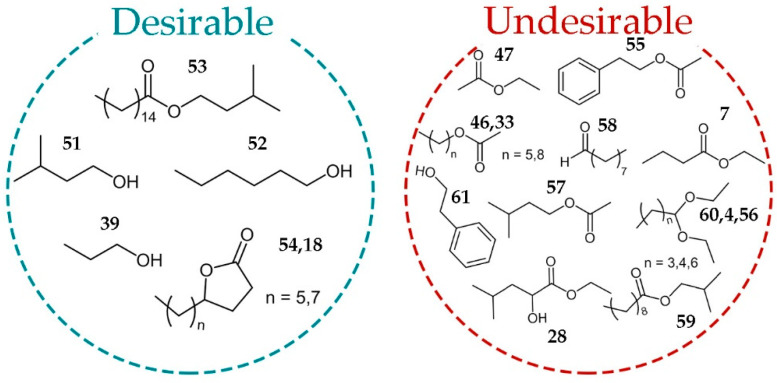
Desirable and undesirable metabolites as determined from correlation of brandies profiles with their sensory properties.

**Table 1 molecules-26-01391-t001:** Parameters of the multivariate analysis models.

Model No.	Model Name	No. of Samples	No. of Components	R^2^	Q^2 1^	*p* (CV-ANOVA)	F (CV-ANOVA)
M1	PCA	108	13	0.831	0.540	-	-
M2	OPLS-DA cultivars CR/PZ ^2^	72	1 + 2	0.991	0.985	<1 × 10^−6^	710
M3	OPLS-DA cultivars TR/PZ	72	1 + 2	0.988	0.981	<1 × 10^−6^	567
M4	OPLS-DA pH of plum mash	108	1 + 4	0.947	0.914	<1 × 10^−6^	103
M5	OPLS-DA natural/selected yeast	108	1 + 4	0.966	0.943	<1 × 10^−6^	159
M6	OPLS-DA with/without Beta	72	1 + 8	0.983	0.907	<1 × 10^−6^	29
M7	OPLS-DA with/without Cuvee	72	1 + 0	0.228	-0.091	1	0
M8	OPLS sensory evaluation	108	1 + 7	0.923	0.809	<1 × 10^−6^	24

^1^ Q^2^ values were determined by 7-fold internal cross-validation. Only the model related to Lallzyme CUVÉE BLANC™ addition with R^2^ around 0.2 and negative Q^2^ value cannot be considered valid for further analysis. ^2^ CR—Crvena ranka; PZ—Požegača; TR—Trnovača.

## Data Availability

Not applicable.

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
