# Peer review of "GC-FID-MS Based Metabolomics to Access Plum Brandy Quality"

_molecules, 2021, doi:10.3390/molecules26051391_

Round 1
Reviewer 1 Report
The authors studied plum brandy metabolomics by using 108 plum brandies originated from three plum cultivars from Serbia. The main volatiles and aroma compounds resulting from the 108 types, that were fermented using different conditions were used to characterize the brandy aroma. Generally, I read the article with interest, I found it well written and organized. However, I have some minor comments, questions and suggestions listed below.
Line 38 – please delete one “are”
Please mention at the end of introduction what is the originality of the article.
Lines 302-307 – please mention the purity of used standards.
Line 308 Please entitled the subheading 3.2. “Plum Sample Collection” or “Fruit Sample Collection”, but not “Plant Sample Collection”.
Is something that I cannot understand about the preparation of the samples and I don't find out how 108 types resulted from the 12 experiments. What were the different fermentation conditions and how many?
Lines 167-168 the authors wrote “bandy samples with lower grades were clearly separated from those with higher grades (Figure 1(h)).” In my opinion they were not clearly separated. Indeed, those with 15 and 16 grades were clearly separated, those with 17 present an attempt to separate, while those with 17.5 to 18.5 are all together. However, try to rephrase it a bit.
I would have liked to see the whole table with the volatile organic compounds detected as supplementary material. The authors might consider presenting this table.
Author Response
Reviewer:
Line 38 – please delete one “are”
Respond: Suggestion was acknowledged and text was changed.
Reviewer: Please mention at the end of introduction what is the originality of the article.
Respond: Lines 105-113. Suggestion was acknowledged and text was changed.
Reviewer: Lines 302-307 – please mention the purity of used standards.
Respond: Suggestion was acknowledged and text was changed.
Reviewer: Line 308 Please entitled the subheading 3.2. “Plum Sample Collection” or “Fruit Sample Collection”, but not “Plant Sample Collection”.
Respond: Suggestion was acknowledged and text was changed.
Reviewer: Is something that I cannot understand about the preparation of the samples and I don't find out how 108 types resulted from the 12 experiments. What were the different fermentation conditions and how many?
Respond: The authors especially appreciate this reviewer comment. Table S1 was added in the Supplementary to explain 12 experiments x 3 cultivars x 3 replicates.
Reviewer: Lines 167-168 the authors wrote “bandy samples with lower grades were clearly separated from those with higher grades (Figure 1(h)).” In my opinion they were not clearly separated. Indeed, those with 15 and 16 grades were clearly separated, those with 17 present an attempt to separate, while those with 17.5 to 18.5 are all together. However, try to rephrase it a bit.
Respond: Lines 193-195. Suggestion was acknowledged and text was changed.
Reviewer: I would have liked to see the whole table with the volatile organic compounds detected as supplementary material. The authors might consider presenting this table.
Respond: Suggestion was acknowledged and tables were included.
Reviewer 2 Report
Journal: Molecules
Title: Plum brandy metabolomics
Authors: Stefan Ivanović, Katarina Simić, Vele Tešević, Ljubodrag Vujisić, Marko Ljekočević and Dejan Gođevac
This is a nice work about plum brandy classification extended with advanced level of chemometric analysis. Figure 3 is particularly useful to visualize the mains findings (metabolites). On the other hand, Figure 2 is not so expressive.
As PLS is a supervised technique several classifications can be carried out, I miss the confusion matrices, the performance metrics for classifications. I do not see the necessity of OPLS, at least the axes on Figure 1c-1h does not infer this technique.
Table 1 is a bit confusing. The determination of Q^2 is not given, though there are many ways to determine it. See e.g. Consonni, V., Ballabio, D., Todeschini, R. Evaluation of model predictive ability by external validation techniques, Journal of Chemometrics Volume 24, Issue 3-4, March 2010, Pages 194-201
DOI: 10.1002/cem.1290
A scientific publication should be fully reproducible. The validation of model 1 cannot satisfactory. Recommended reading: Tóth, G., Bodai, Z., Héberger, K. Estimation of influential points in any data set from coefficient of determination and its leave-one-out cross-validated counterpart Journal of Computer-Aided Molecular Design Volume 27, Issue 10, October 2013, Pages 837-844
DOI: 10.1007/s10822-013-9680-4
i.e. some influential points, outliers are suspected.
The column of “p (CV-ANOVA)” is illusory, 0 means significant difference, and 10to-21 measn the same (and the validity of this value is questionable because of the digital precision and imprecision of the calculation algorithm). Similarly, the F column is illusory with six value digits of precision.
Although I find the experiments and the obtained data are valuable, but, on the contrary the conclusions are weak. The first sentence is not a concluding remark. The second one is trivial (“The GC-FID-MSD-based metabolomics …be a very reliable”)
Who has ever doubted it? The third one is better still weak. “Correlation of brandy GC-FID-MSD profiles with their sensory properties achieved by OPLS multivariate analysis” Very well, bat were the results?
The references are over weighed to Serbian authors without justification.
I miss the next and similar references:
Bajer, T., Hill, M., Ventura, K., Bajerová, P. Authentification of fruit spirits using HS-SPME/GC-FID and OPLS methods Scientific Reports Volume 10, Issue 1, 1 December 2020, Article number 18965
DOI: 10.1038/s41598-020-75939-0
Jakubíková, M., Sádecká, J., Hroboňová, K. Classification of plum brandies based on phenol and anisole compounds using HPLC European Food Research and Technology Volume 245, Issue 8, 1 August 2019, Pages 1709-1717
DOI: 10.1007/s00217-019-03291-3
Ilea, M., Fitiu, A., Vac, S.C. Studies on technological characteristics of marin brandy production for certification as a local and organic product
Journal of Environmental Protection and Ecology Volume 20, Issue 1, 2019, Pages 337-347
Jakubíková, M., Sádecká, J., Kleinová, A. On the use of the fluorescence, ultraviolet–visible and near infrared spectroscopy with chemometrics for the discrimination between plum brandies of different varietal origins
Food Chemistry Volume 239, 15 January 2018, Pages 889-897
DOI: 10.1016/j.foodchem.2017.07.008
Vyviurska, O., Matura, F., Furdíková, K., Špánik, I. Volatile fingerprinting of the plum brandies produced from different fruit varieties Journal of Food Science and Technology Volume 54, Issue 13, 1 December 2017, Pages 4284-4301
DOI: 10.1007/s13197-017-2900-5
Pielech-Przybylska, K., Balcerek, M., Nowak, A., Patelski, P., Dziekońska-Kubczak, U. Influence of yeast on the yield of fermentation and volatile profile of ‘Węgierka Zwykła’ plum distillates Journal of the Institute of Brewing Volume 122, Issue 4, October 2016, Pages 612-623
DOI: 10.1002/jib.374
Satora, P., Tuszyński, T. Chemical characteristics of Śliwowica Ła̧cka and other plum brandies Journal of the Science of Food and Agriculture Volume 88, Issue 1, January 2008, Pages 167-174
DOI: 10.1002/jsfa.3067
Winterová, R., Mikulíková, R., Mazáč, J., Havelec, P. Assessment of the authenticity of fruit spirits by gas chromatography and stable isotope ratio analyses, Czech Journal of Food Sciences Volume 26, Issue 5, 2008, Pages 368-375
DOI: 10.17221/1610-cjfs
And many others. Naturally I cannot prescribe what to cite, but I suggest a serious overview, the following actions are strongly recommended: selection of the best papers, reduction of the Serbian sources to an absolute minimum, making comparison with earlier findings, etc.
Were the data scaled to unit variance orunit standard deviation?
There are many more minor errors, but first the more serious ones should be corrected.
February 11 / 2021 referee
Author Response
Reviewer: This is a nice work about plum brandy classification extended with advanced level of chemometric analysis. Figure 3 is particularly useful to visualize the mains findings (metabolites). On the other hand, Figure 2 is not so expressive.
Respond: Lines 210-211. Suggestion was acknowledged and text was included in Figure 2 caption for better expressiveness.
Reviewer: As PLS is a supervised technique several classifications can be carried out, I miss the confusion matrices, the performance metrics for classifications. I do not see the necessity of OPLS, at least the axes on Figure 1c-1h does not infer this technique.
Respond: Suggestion was acknowledged and table S4 was included. We used SIMCA software, and the misclassification tables (the analogs of confusion matrices) of the OPLS-DA models are now given.
As stated in the manuscript: “An additional advantage of the orthogonal model is the facilitated interpretation due to separation of the systematic variation of the variables into two parts: one linearly related to class information and one orthogonal to the class information [Wiklund et al. Visualization of GC/TOF-MS-based metabolomics data for identification of biochemically interesting compounds using OPLS class models. Anal. Chem. 2008, 80, 115–122]. Thus, OPLS-DA is suitable for finding variables having the greatest discriminatory power between classes.”
This resulted in models with much improved interpretability comparing to PLS. We tried corresponding PLS-DA and PLS models, but in such models the predictive information of Y in X is not concentrated in one component. Using orthogonal models, the variation in X which is related to Y is pushed in the first (predictive) component, while the variation in X which is unrelated to Y is pushed in orthogonal components (second, third…)
Reviewer: Table 1 is a bit confusing. The determination of Q^2 is not given, though there are many ways to determine it.
Respond: Line 142. Suggestion was acknowledged and text was included in the footer of Table 1
Reviewer: A scientific publication should be fully reproducible. The validation of model 1 cannot satisfactory. i.e. some influential points, outliers are suspected.
Respond: Line 135-137. Suggestion was acknowledged and text was included. The cross-validation procedure has become golden standard in MVA, which is incorporated in different forms in most commercial software. However, cross-validation is implemented differently in different software packages, which may cause many confusions when comparing models developed by different packages. The model 1 (PCA) is also validated by 7-fold cross-validation, following the procedure described for SIMCA software.
No outliers in PC1/PC2 score plot (Figure 1(a)) observed.
Reviewer: The column of “p (CV-ANOVA)” is illusory, 0 means significant difference, and 10to-21 measn the same (and the validity of this value is questionable because of the digital precision and imprecision of the calculation algorithm). Similarly, the F column is illusory with six value digits of precision.
Respond: Suggestion was acknowledged and Table 1 was changed accordingly. Regarding the p-value issue, I received the answer from software engineer: “p-value is stored in a 32 bit value, the smallest possible value it can have without being 0 is 1.175494351e-38. But the rounding errors are much much bigger due to the precision of the values, it’s about 6 decimals for 32 bit floats so < 1e-6 is an good answer.”
Reviewer: Although I find the experiments and the obtained data are valuable, but, on the contrary the conclusions are weak. The first sentence is not a concluding remark. The second one is trivial (“The GC-FID-MSD-based metabolomics …be a very reliable”) Who has ever doubted it? The third one is better still weak. “Correlation of brandy GC-FID-MSD profiles with their sensory properties achieved by OPLS multivariate analysis” Very well, bat were the results?
Respond: The authors especially appreciate this reviewer comment. Lines 442-460. Suggestion was acknowledged and text was changed.
Reviewer: The references are over weighed to Serbian authors without justification. Naturally I cannot prescribe what to cite, but I suggest a serious overview, the following actions are strongly recommended: selection of the best papers, reduction of the Serbian sources to an absolute minimum, making comparison with earlier findings, etc.
Respond: Suggestion was acknowledged and text was changed (Lines 74-76; 89-101; 219-225), the references included (Lines 503-505; 511-518; 525-527.) Some of the references are with Serbian authors, but this is because these authors are deeply engaged in brandy investigation for years. Please note that the similarity in the author's surnames may lead to a wrong conclusion. There are many authors cited who are Croats or Bosnians.
Round 2
Reviewer 2 Report
The authors have revised their manuscript according to the reviewer's suggestions. I could imagine a more detailed comparison with literature date in the discussions, but this is not mandatory.
Author Response
Suggestion was acknowledged and (Lines 251-253)
Spelling and grammar errors are corrected throughout the text.